# Improving Viewpoint-Independent Object-Centric Representations through Active Viewpoint Selection

**Yinxuan Huang, Chengmin Gao, Bin Li**[*]**, Xiangyang Xue**[*]
Shanghai Key Laboratory of Intelligent Information Processing
School of Computer Science, Fudan University
yxhuang22@m.fudan.edu.cn, {19210240036, libin, xyxue}@fudan.edu.cn

## Abstract

Given the complexities inherent in visual scenes, such as object occlusion, a comprehensive understanding often requires observation from multiple viewpoints. Existing multi-viewpoint object-centric learning methods typically employ random or sequential viewpoint selection strategies. While applicable across various scenes, these strategies may not always be ideal, as certain scenes could benefit more from specific viewpoints. To address this limitation, we propose a novel active viewpoint selection strategy. This strategy predicts images from unknown viewpoints based on information from observation images for each scene. It then compares the object-centric representations extracted from both viewpoints and selects the unknown viewpoint with the largest disparity, indicating the greatest gain in information, as the next observation viewpoint. Through experiments on various datasets, we demonstrate the effectiveness of our active viewpoint selection strategy, significantly enhancing segmentation and reconstruction performance compared to random viewpoint selection. Moreover, our method can accurately predict images from unknown viewpoints.

## 1 Introduction

Humans typically perceive a visual scene as combining various visual concepts, including objects, backgrounds, and their basic parts. This object-level perception allows for a better understanding of diverse environments that consist of multiple objects and backgrounds. Similarly, object-centric learning focuses on the object-level representation of images or videos instead of modeling the entire scene directly [1]. Such object-centric representations prove to be more versatile for a range of visual tasks, such as visual scene understanding [2], visual reasoning [3], and causal inference [4].

Given the inherent complexities of visual scenes, such as object occlusion, capturing the entire scene comprehensively from a single viewpoint is often challenging. Therefore, observing the scene from multiple viewpoints becomes essential to achieve a deeper understanding. In recent years, various object-centric learning methods have emerged to address multi-viewpoint learning without object-level supervision, including MulMON [5], ROOTS [6], SIMONe [7], TC-VDP [8], and OCLOC [9, 10]. MulMON and ROOTS leverage provided viewpoint annotations to learn viewpoint-independent object-centric representations, relying heavily on these annotations. In contrast, SIMONe and TC-VDP do not require explicit viewpoint annotations, instead assuming temporal relationships among viewpoints within the same scene and using frame indexes for inference. Unlike these approaches, OCLOC operates in a fully unsupervised manner, where viewpoints are both unknown and unrelated.

---

[*]Corresponding author.

38th Conference on Neural Information Processing Systems (NeurIPS 2024).

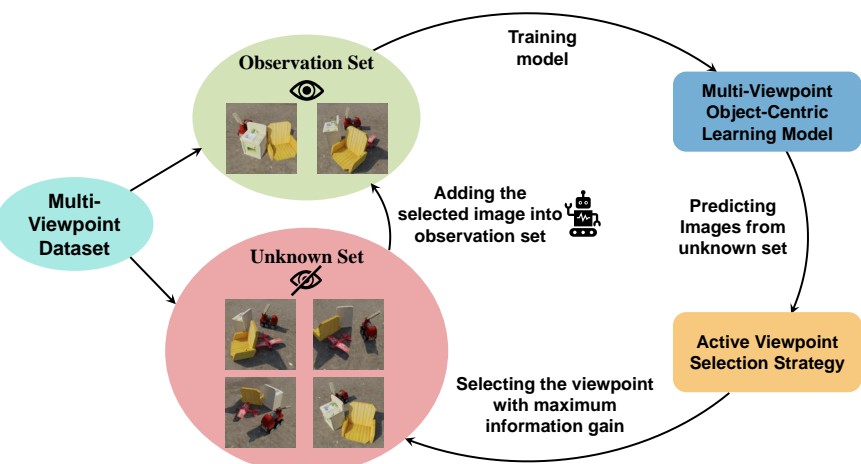

Figure 1: **Active viewpoint selection framework.** Our proposed method iteratively selects viewpoints from the unknown set to form a small yet informative observation set, enabling effective training with fewer images. The active viewpoint selection strategy evaluates the information gain of the unknown viewpoints using the predicted images and selects the viewpoint with the maximum information gain as the next observation. The real image of the selected viewpoint is then added to the observation set, and this process continues until the observation set reaches a predefined size.

However, these multi-viewpoint object-centric learning methods generally utilize images from either random or sequential viewpoints as input. While these viewpoint selection strategies can be applied to various scenes, they are not always ideal. Typically, capturing a full understanding of the visual scene requires images from multiple viewpoints (e.g., 6–8), which may introduce inefficiencies. Notably, certain scenes may be more sensitive to information from specific viewpoints, meaning a generic selection strategy could lead to redundancy or overlook essential scene details.

Moreover, despite significant advancements in unsupervised object segmentation, the generative capabilities of multi-viewpoint methods still require improvement, largely due to the limitations of their mixture-based decoders. Recently, LSD [11] and SlotDiffusion [12] have combined object-centric learning models with diffusion-based slot decoders, leveraging the powerful image generation capabilities of the diffusion model [13, 14] to achieve high-quality slot-to-image decoding. However, neither approach is suitable for multi-viewpoint scenes, as one is designed for single images and the other for videos. The object-centric representations they learn are specific to individual images, rather than being scene-specific or viewpoint-independent. While these methods support certain image generation and editing tasks, they lack the ability to synthesize novel views.

To address the limitation mentioned above, we propose AVS, a novel multi-viewpoint object-centric learning model with an active viewpoint selection strategy. AVS optimizes viewpoint selection and enhances image decoding quality through the integration of a diffusion model. As illustrated in Figure 1, images are divided into an observation set and an unknown set. AVS learns viewpoint-independent object-centric representations from the observation set and predicts images in the unknown set. It then extracts object-centric representations from the unknown viewpoints and compares them with those from the observation set. The unknown viewpoint with the largest disparity, indicating maximum information gain, is selected as the next observation viewpoint. Repeating this process allows the model to refine viewpoint-independent object-centric representations by actively selecting the most informative viewpoints.

To validate the advantages of our active viewpoint selection strategy, we conducted performance comparison experiments between the active selection strategy and the random selection strategy using the same model architecture. The experimental results indicate that active viewpoint selection achieves better segmentation performance than random selection with the same number of viewpoints. Furthermore, we compare the performance of our model with other multi-viewpoint object-centric learning methods. The results demonstrate that our model achieves superior segmentation performance and outstanding generation capability. Additionally, our method can predict images from unknown viewpoints and supports novel viewpoint synthesis.

In summary, our contributions are as follows: 1) We propose AVS, a multi-viewpoint object-centric learning model with an active viewpoint selection strategy, which demonstrates improved performance in unsupervised object segmentation and image generation compared to baseline methods. 2) Our active viewpoint selection strategy significantly enhances viewpoint-independent object-centric representations, enabling the model to better understand and perceive visual scenes. 3) Our model can predict images from unknown viewpoints and generate images from novel viewpoints.

## 2 Related Work

Object-centric learning aims to learn compositional representations of visual scenes by decomposing them into a set of object-level feature vectors. Existing approaches can be categorized along two main dimensions: the type of input data and the type of decoder used.

**Based on the type of input data.** Object-centric learning methods can be categorized into single-image-based, video-based, and multi-viewpoint-based approaches depending on the type of input data. Many classical object-centric methods, such as N-EM [15], AIR [2], and Slot Attention [16], learn compositional representations from a single image, establishing mechanisms to infer object-centric representations that form the basis for video- and multi-viewpoint-based approaches. Video-based methods, such as SAVi [17], utilize multi-frame videos as input, often leveraging temporal cues to model object motion and relationships and to maintain object identities across frames. Multi-viewpoint-based methods, such as MulMON [5], SIMONe [7], and OCLOC [9, 10], use multi-viewpoint images as input and sometimes rely on viewpoint annotations. These approaches typically model viewpoint representations individually and learn viewpoint-independent object representations for each scene. Unlike video-based methods, where video frames are continuous, the viewpoints in multi-viewpoint-based methods may be independent and unrelated. In contrast to previous approaches, where observation viewpoints are predefined or selected with a generic strategy (e.g., random or sequential), we propose an approach that actively selects viewpoints based on specific scene information.

**Based on the type of decoder.** Object-centric learning methods can also be categorized by the type of decoder used to reconstruct the input image: mixture-based decoders, transformer-based decoders, and diffusion-based decoders. Mixture-based decoders process each object representation individually, combining results through weighted summation to obtain the final reconstruction. While this approach promotes independence between object representations, it demands significant computational time and memory, often resulting in blurred, less detailed images. Transformer-based decoders, proposed by SLATE [18] and STEVE [19], employ transformer architecture to decode the set of object representations autoregressively, yielding more detailed reconstructions and effectively segmenting naturalistic images and videos. However, in more complex scenes, the slot-to-image reconstruction quality of transformer-based decoders remains limited. Diffusion-based decoders, introduced by LSD [11] and SlotDiffusion [12], leverage the powerful image generation capabilities of diffusion models to produce diverse and realistic images. However, current diffusion-based methods are limited to single images and videos, without support for multi-viewpoint scenes. To overcome this limitation, we propose a multi-viewpoint model with a diffusion-based decoder, combining the powerful image generation capabilities of diffusion models with the ability to generate images from novel viewpoints.

## 3 Method

For a visual scene observed from $V$ viewpoints, let $\mathcal{U} = \{(\boldsymbol{x}_1, \boldsymbol{c}_1), ..., (\boldsymbol{x}_V, \boldsymbol{c}_V)\}$ represent the universal set of multi-viewpoint images paired with their corresponding viewpoint timesteps. We partition $\mathcal{U}$ into two mutually exclusive sets: $\mathcal{O}$ and $\mathcal{P}$, where $\mathcal{O} = \{(\boldsymbol{x}_{\tau_1}, \boldsymbol{c}_{\tau_1}), ..., (\boldsymbol{x}_{\tau_M}, \boldsymbol{c}_{\tau_M})\}$ and $\mathcal{P} = \{(\boldsymbol{x}_{\upsilon_1}, \boldsymbol{c}_{\upsilon_1}), ..., (\boldsymbol{x}_{\upsilon_L}, \boldsymbol{c}_{\upsilon_L})\}$ denote the sets of observation and unknown viewpoints, respectively. Here, $\tau$ and $\upsilon$ are complementary increasing subsequences of $[1, ..., V]$ with lengths $M$ and $L$.

Figure 2 provides an overview of our model. Our active viewpoint selection strategy aims to learn *viewpoint-independent object-centric representations* that capture the 3D structure and fine-grained textures by optimally selecting viewpoints from $\mathcal{P}$. Building upon Latent Slot Diffusion [11], the key components to achieving this include: 1) learning viewpoint-independent object-centric

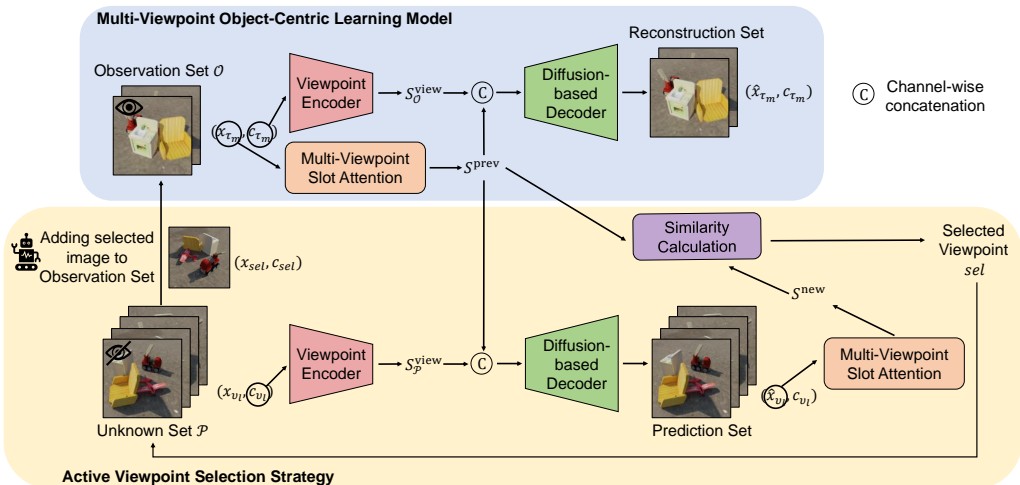

Figure 2: **Model architecture overview.** Given the observation set $\mathcal{O}$, model learns viewpoint-independent object-centric representations $S^{\text{prev}}$ from Multi-Viewpoint Slot Attention and viewpoint representations $S^{view}_{\mathcal{O}}$ from Viewpoint Encoder. These representations are concatenated and input into the Diffusion-base Decoder to reconstruct the observation set. For the unknown set $\mathcal{P}$, model obtains viewpoint representations $S^{view}_{\mathcal{P}}$ from Viewpoint Encoder. $S^{\text{prev}}$ and $S^{view}_{\mathcal{P}}$ are concatenated and input into the Diffusion-base Decoder to predict images. The object representations $S^{\text{new}}$ are obtained from the predicted image and compared with $S^{\text{prev}}$ to evaluate the information gain of the unknown viewpoint. The viewpoint with the maximum information gain is selected and its corresponding real image $\boldsymbol{x}_{sel}$ is added to the observation set.

representations (slots) from multiple viewpoints through multi-viewpoint slot attention and slot-conditioned diffusion (detailed in Section 3.1), 2) selecting the unknown viewpoint by minimizing slot similarity through generative iterations (discussed in Section 3.2), and 3) training details in Section 3.3.

## 3.1 Multi-Viewpoint Latent Slot Diffusion

### 3.1.1 Background: Latent Slot Diffusion

Latent Slot Diffusion (LSD) [11] consists of two main components: the Object-Centric Encoder and the Slot-Conditioned Diffusion Decoder in the latent space.

**Object-Centric Encoder.** The Object-Centric Encoder utilizes Slot Attention [16] to encode an input image $\boldsymbol{x} \in \mathbb{R}^{H \times W \times C}$ into features and aggregate this information into a collection of $K$ slots $\mathcal{S} \in \mathbb{R}^{K \times D}$, where $D$ represents the dimensionality of each slot, capturing the semantic entities of the image. Let $\boldsymbol{h} \in \mathbb{R}^{N \times D_{\text{enc}}}$ be the flattened feature map obtained through the encoder $f^{\phi}_{\text{enc}}(\boldsymbol{x})$. The core of Slot Attention involves iteratively updating $K$ slots $\tilde{\mathcal{S}}$, initialized by sampling from a Gaussian distribution with learnable parameters, using iterative attention: $\mathcal{S} = f^{\phi}_{\text{SA}}(\tilde{\mathcal{S}}, \boldsymbol{h})$. The function $f^{\phi}_{\text{SA}}$ clusters features to capture $K$ soft regions and employs a Gated Recurrent Unit (GRU) [20] to update the slots. We extend Slot Attention to a multi-viewpoint version to enhance slot representations, as detailed in Section 3.1.2.

**Slot-Conditioned Diffusion Decoder.** Similar to Stable Diffusion (SD) [21], LSD aims to learn the prior $p(\boldsymbol{z}_0 \mid \mathcal{S})$ conditioned on the slots $\mathcal{S}$, leveraging the generative capabilities of Diffusion Models (DMs) [13, 14]. Here, $\boldsymbol{z}_0$ represents the latent representation of the image obtained through a pre-trained encoder $\boldsymbol{z}_0 = f^{\text{SD}}_{\text{enc}}(\boldsymbol{x})$. DMs apply a variance-preserving Markov process to $\boldsymbol{z}_0$, progressively increasing noise levels to create various noisy latent representations:

$$\boldsymbol{z}_t = \sqrt{\bar{\alpha}_t}\boldsymbol{z}_0 + \sqrt{1 - \bar{\alpha}_t}\boldsymbol{\epsilon}_t, \quad \boldsymbol{\epsilon}_t \sim \mathcal{N}(\boldsymbol{0}, \boldsymbol{I}) \tag{1}$$

where $t \in \{1, ..., T\}$ represents the timesteps in the Markov process, $\bar{\alpha}_t = \prod_{i=1}^{t}(1 - \beta_i)$, and $\beta_t$ is a monotonically increasing weighting schedule. LSD aims to predict $\boldsymbol{\epsilon}_t$ at each timestep $t$ using a

denoising network that incorporates position encoding and is conditioned on $\boldsymbol{z}_t$, the timestep $t$, and the slots $\mathcal{S}(\boldsymbol{x}; \phi)$, i.e., $\hat{\boldsymbol{\epsilon}}_t = \boldsymbol{\epsilon}_\theta(\boldsymbol{z}_t, t, \mathcal{S}(\boldsymbol{x}; \phi))$. LSD trains DMs by minimizing the expected mean squared error (MSE) between $\hat{\boldsymbol{\epsilon}}_t$ and $\boldsymbol{\epsilon}_t$:

$$\mathcal{L}(\theta, \phi) = \mathbb{E}_{t \sim \text{Uniform}(1,\dots,T), \boldsymbol{\epsilon}_t \sim \mathcal{N}(\mathbf{0}, \boldsymbol{I})} \left[ \lambda_t \| \boldsymbol{\epsilon}_t - \hat{\boldsymbol{\epsilon}}_t \|_2^2 \right] \tag{2}$$

Here, $\lambda_t$ is a hyperparameter that weights the loss at timestep $t$. After training, the model can gradually denoise from $\boldsymbol{z}_T \sim \mathcal{N}(\mathbf{0}, \boldsymbol{I})$ to $\hat{\boldsymbol{z}}_0$ using a fast diffusion sampler [22, 23], and then decode $\hat{\boldsymbol{z}}_0$ back into the image space $\hat{\boldsymbol{x}} = f_{\text{dec}}^{\text{SD}}(\hat{\boldsymbol{z}}_0)$.

### 3.1.2 Learning Slots from Multiple Viewpoints

Capturing complete object-centric representations is challenging due to occlusion, complex backgrounds, and diverse object categories. To address this, we propose a Multi-Viewpoint Slot Attention Algorithm (see Algorithm 1) for learning comprehensive and viewpoint-independent slots.

Firstly, we replace $f_{\text{enc}}^\phi$ with a frozen DINO ViT [24] to enhance feature extraction. Next, we encode $V$ viewpoint timesteps into viewpoint representations $\mathcal{S}^{\text{view}}$ using a viewpoint encoder $f_{\text{enc}}^{\text{view}}$. A key aspect of Multi-Viewpoint Slot Attention is that $\mathcal{S}^{\text{view}}$ provides viewpoint information during Slot Attention iterations. In each iteration, it averages the updated slots across all viewpoints, thus making the object representations $\mathcal{S}$ viewpoint-independent. For more details, refer to Lines 7-11 in Algorithm 1.

---

**Algorithm 1:** Multi-Viewpoint Slot Attention

**Input:** $\mathcal{X}$: multi-viewpoint images; $K$: maximum number of objects; $T$: number of iterations; $D$: dimension of slots; $\mathcal{S}^{\text{view}}$: viewpoint representations; $\mathcal{S}$: object representations (slots)

1 `//` $\hat{\boldsymbol{\mu}}, \hat{\boldsymbol{\sigma}}$: `learnable parameters;` $k, q, v$: `linear projections for attention`

2 **Function** SlotAttn$(\mathcal{X}, K, T, D, \mathcal{S}^{view}, \mathcal{S} = \varnothing)$:

3      $\boldsymbol{h}_m = \text{DINO}(\boldsymbol{x}_m) \in \mathbb{R}^{N \times D_{\text{enc}}}, \quad \forall \boldsymbol{x}_m \in \mathcal{X}$

4      **if** $\mathcal{S} = \varnothing$ **then**

5          $\mathcal{S} \sim \mathcal{N}\big(\hat{\boldsymbol{\mu}}, \text{diag}(\hat{\boldsymbol{\sigma}})\big) \in \mathbb{R}^{K \times D}$

6      **for** $t \leftarrow 1$ **to** $T, \forall \boldsymbol{x}_m \in \mathcal{X}$ **do**

7          $\mathcal{S}_m^{\text{full}} = [\mathcal{S}_m^{\text{view}}, \mathcal{S}]$

8          $\boldsymbol{A}_m = \text{Softmax}\left(\frac{1}{\sqrt{D}} k(\boldsymbol{h}_m) \cdot q(\mathcal{S}_m^{\text{full}})^\top, \text{axis='slots'}\right)$

9          $\boldsymbol{U}_m = \text{WeightedMean}\left(\text{weights} = \boldsymbol{A}_m, \text{values} = v(\boldsymbol{h}_m)\right)$

10         $\tilde{\mathcal{S}}_m^{\text{full}} = \text{GRU}\left(\text{state} = \mathcal{S}_m^{\text{full}}, \text{inputs} = \boldsymbol{U}_m\right), \quad \left[\boldsymbol{S}_m^{\text{view}}, \boldsymbol{S}_m^{\text{attr}}\right] \overset{\text{split}}{=} \tilde{\mathcal{S}}_m^{\text{full}}$

11         $\mathcal{S} = \text{Mean}\left(\boldsymbol{S}_{1:|\mathcal{X}|}^{\text{attr}}, \text{axis='viewpoint'}\right)$

12      **return** $\mathcal{S}$

---

## 3.2 Active Viewpoint Selection

To enhance viewpoint-independent object-centric representations learned from multi-viewpoint images, we propose an active viewpoint selection strategy that optimizes representation learning by strategically selecting a small, informative subset of viewpoints. As outlined in Algorithm 2, our proposed strategy iteratively selects the next observation viewpoint from the unknown set $\mathcal{P}$ to maximize information gain until the observation set $\mathcal{O}$ reaches a predefined maximum size $M$. In each iteration, the model predicts images from unknown viewpoints based on the current observation set and estimates the information gain for each candidate viewpoint. The viewpoint with the highest estimated gain is then selected, and its actual image is added to the observation set. Specifically, the algorithm proceeds as follows:

**Step 1. Initialization (Lines 2-5)**: An image is randomly selected from the universal set $\mathcal{U}$ to form the initial observation set $\mathcal{O}$, and initial object representations $\mathcal{S}^{\text{prev}}$ are computed based on this single viewpoint.

**Step 2. Viewpoint Prediction and Selection Loop (Lines 8-14)**: This step, referred to as the *inner loop*, iterates through each candidate viewpoint in the unknown set $\mathcal{P}$: 1) Concatenate $\mathcal{S}^{\text{prev}}$ with

the viewpoint representations $\mathcal{S}_i^{\text{view}}$ of each candidate viewpoint to generate a reconstructed image $\hat{\boldsymbol{x}}_i$ using a pre-trained slot-conditioned diffusion decoder. 2) Update the object representations $\mathcal{S}^{\text{new}}$ based on the predicted image $\hat{\boldsymbol{x}}_i$ and the observation set $\mathcal{O}$. 3) Calculate the cosine similarity between $\mathcal{S}^{\text{prev}}$ and $\mathcal{S}^{\text{new}}$ to assess the information gain of each candidate viewpoint:

$$\text{Sim}(\mathcal{S}^{\text{prev}}, \mathcal{S}^{\text{new}}) = \sum\nolimits_{k=1}^{K} \text{CosSim}\left(\mathcal{S}_k^{\text{prev}}, \mathcal{S}_k^{\text{new}}\right) = \sum\nolimits_{k=1}^{K} \frac{\mathcal{S}_k^{\text{prev}} \cdot \mathcal{S}_k^{\text{new}}}{\|\mathcal{S}_k^{\text{prev}}\|_2 \cdot \|\mathcal{S}_k^{\text{new}}\|_2} \tag{3}$$

4) Select the candidate viewpoint with the maximum information gain as the next observation viewpoint, and add the corresponding real image $x_{sel}$ to the observation set $\mathcal{O}$.

**Step 3. Iterative Viewpoint Selection (Lines 6-16)**: This process forms the *outer loop*. The viewpoint selection loop is repeated until a total of $M$ images have been observed, at which point the final observation viewpoint set $\mathcal{O}$ is returned.

The active viewpoint selection process depends on a pre-trained slot-conditioned diffusion model, as without it, the generation results from the fast sampler may not effectively support selection. Additionally, since the selection process does not involve gradient calculation, the fast sampler can operate with fewer steps (e.g., 5 to 10), which quickly provides rough feature references for Slot Attention without compromising selection capability.

---

**Algorithm 2:** Active Viewpoint Selection Algorithm

**Data:** $\mathcal{U} = \left\{ (\boldsymbol{x}_1, \boldsymbol{c}_1), ..., (\boldsymbol{x}_V, \boldsymbol{c}_V) \right\}$
**Input:** $M$: the maximum number of images can be selected from $\mathcal{U}$ for observation
**Output:** $\mathcal{O}$: observation viewpoint set; $\mathcal{P}$: unknown viewpoint set; $\mathcal{S}$: object representations
1 // Lines 1-16 are executed without gradient calculations
2 $(\boldsymbol{x}_0, \boldsymbol{c}_0) = \text{UniformSampling}(\mathcal{U}), \mathcal{O} = (\boldsymbol{x}_0, \boldsymbol{c}_0), \mathcal{P} = \mathcal{U} - \mathcal{O}$
3 $\mathcal{S}_v^{\text{view}} = f_{\text{enc}}^{\text{view}}(\boldsymbol{c}_v), \quad \forall 1 \le v \le V$
4 $\mathcal{S}_{\mathcal{O}}^{\text{view}} = \text{SelectByIndex}(\mathcal{S}^{\text{view}}, \mathcal{O})$
5 $\mathcal{S}^{\text{prev}} = \text{SlotAttn}(\mathcal{O}, K, T, D, \mathcal{S}_{\mathcal{O}}^{view})$
6 **while** $|\mathcal{O}| < M$ **do**
7 $\quad sel = \text{None}, score = \inf, \mathcal{S}^{\text{upd}} = \text{None}$
8 $\quad$ **for** $(x_i, c_i) \in \mathcal{P}$ **do**
9 $\quad\quad \mathcal{S}^{\text{full}} = [\mathcal{S}_i^{\text{view}}, \mathcal{S}^{\text{prev}}]$
10 $\quad\quad \hat{\boldsymbol{x}}_i = f_{\text{dec}}^{\text{SD}}\left(\text{FastSampler}(\boldsymbol{\epsilon}, \mathcal{S}^{\text{full}}; \theta)\right), \quad \boldsymbol{\epsilon} \sim \mathcal{N}(\boldsymbol{0}, \boldsymbol{I})$
11 $\quad\quad \mathcal{O}' = \mathcal{O} \cup \left\{ (\hat{\boldsymbol{x}}_i, \boldsymbol{c}_i) \right\}, \mathcal{S}_{\mathcal{O}'}^{\text{view}} = \mathcal{S}_{\mathcal{O}}^{\text{view}} \cup \left\{ \mathcal{S}_i^{\text{view}} \right\}$
12 $\quad\quad \mathcal{S}^{\text{new}} = \text{SlotAttn}(\mathcal{O}', K, T, D, \mathcal{S}_{\mathcal{O}'}^{view}, \mathcal{S} = \mathcal{S}^{prev})$
13 $\quad\quad$ **if** $Sim(\mathcal{S}^{prev}, \mathcal{S}^{new}) < score$ **then**
14 $\quad\quad\quad sel = i, score = \text{Sim}(\mathcal{S}^{\text{prev}}, \mathcal{S}^{\text{new}}), \mathcal{S}^{\text{upd}} = \mathcal{S}^{\text{new}}$
15 $\quad \mathcal{O} = \mathcal{O} \cup \left\{ (\boldsymbol{x}_{sel}, \boldsymbol{c}_{sel}) \right\}, \mathcal{P} = \mathcal{U} - \mathcal{O}$
16 $\quad \mathcal{S}^{\text{prev}} = \mathcal{S}^{\text{upd}}$
17 $\mathcal{S} = \text{SlotAttn}(\mathcal{O}, K, T, D, \mathcal{S}_{\mathcal{O}}^{view})$
18 **return** $\mathcal{O}, \mathcal{P}, \mathcal{S}$;

---

### 3.3 Training

We first pre-train a single-viewpoint latent diffusion model using Eq.(2). To stabilize training across multiple viewpoints, we train a feature decoder $f_{\text{dec}}^{\psi}$ to reconstruct feature vectors $\boldsymbol{h}$ in the initial stage. This helps to properly initialize the slots $\mathcal{S}$ before training the complete model. The loss function is defined as follows:

$$\mathcal{L} = \sum\nolimits_{(\boldsymbol{x}_i, \boldsymbol{c}_i) \in \mathcal{O}} \lambda \|\boldsymbol{\epsilon}_{t(i)} - \hat{\boldsymbol{\epsilon}}_{t(i)}\|_2^2 + (1 - \lambda) \|f_{\text{dec}}^{\psi}(\mathcal{S}) - \boldsymbol{h}_i\|_2^2 \tag{4}$$

where $\lambda$ is a non-decreasing hyperparameter that is adjusted throughout the training process, $t(i)$ denotes the denoising timestep for the $i$th viewpoint $(\boldsymbol{x}_i, \boldsymbol{c}_i)$ from the observation set $\mathcal{O}$, and $\hat{\boldsymbol{\epsilon}}_{t(i)}$ corresponds to $\hat{\boldsymbol{\epsilon}}_t$ as defined in Eq.(2).

# 4 Experiments

To evaluate our proposed model, we focus on four tasks: unsupervised object segmentation, scene reconstruction, compositional generation, and novel viewpoint synthesis. For the first two tasks, we compare our active viewpoint selection strategy with the random viewpoint selection strategy to highlight its superiority. Additionally, we evaluate our model against other multi-viewpoint approaches, including SIMONe [7] and OCLOC [9, 10], as well as the single-image-based method LSD [11]. These comparisons demonstrate the advantages of our model in unsupervised object segmentation and scene reconstruction. For the third task, we showcase our model's ability to generate multi-viewpoint samples and perform viewpoint interpolation, and compare its image generation capabilities with baseline methods. For the fourth task, we demonstrate our model's ability to predict images from unknown viewpoints. Our model not only excels in segmentation and reconstruction for unknown viewpoints but also effectively generates images for novel viewpoints.

**Datasets.** We generated three synthetic multi-object multi-viewpoint datasets, referred to as CLEVR-TEX, GSO, and ShapeNet, to evaluate the performance of our model. These datasets were constructed based on the CLEVRTEX dataset [25], the GSO dataset [26], and the ShapeNet dataset [27], respectively. They were created using the official code provided by CLEVRTEX [25] and Kubric [28]. The image size for all datasets is $128 \times 128$. The total number of viewpoints is 12 for CLEVRTEX and 8 for both GSO and ShapeNet. More details on all datasets can be found in Appendix A.

**Evaluation metrics.** Several evaluation metrics are used to evaluate the performance of different methods from three aspects. 1) Adjusted Rand Index (ARI) [29] and mean Intersection over Union (mIoU) assess the quality of segmentation. To provide a more comprehensive evaluation, we utilize two variants of ARI: ARI-A, which considers both objects and background, and ARI-O, which focuses exclusively on objects. The calculation of ARI and mIoU follows OCLOC [9, 10], accounting for object consistency across viewpoints. Since our method does not model complete object shapes, mIoU is computed based on perceived shapes. 2) Learned Perceptual Image Patch Similarity (LPIPS) [30] measures the quality of reconstruction. LPIPS evaluates differences at the feature level and aligns more closely with human perception. 3) Fréchet Inception Distance (FID) [31] assesses the diversity and quality of generated images.

**Implementation details.** To demonstrate the efficiency of our model, we trained the compared methods using images from 8 viewpoints per scene, while our proposed model was trained with images from only 4 viewpoints. This shows that our model can achieve superior segmentation and reconstruction performance, as well as learn better object-centric representations, even with fewer viewpoints. SIMONe was trained with sequential viewpoints, while OCLOC and LSD were trained with random viewpoints. The training process for our approach differs between the **random** and **active** strategies: **random** involves directly selecting 4 random viewpoints and training with Eq 2, while **active** requires pretraining a single-viewpoint model (by randomly selecting a viewpoint) and then training with Eq 2 following Algorithm 2, where the number of viewpoints in $\mathcal{O}$ is 4. Images from all viewpoints in the test set were used for evaluation, and all experimental results were repeated 3 times. Additional details are provided in Appendix B.

## 4.1 Unsupervised Object Segmentation

We present quantitative experimental results in Table 1 and visualize qualitative results of unsupervised object segmentation in Figure 3. Our model demonstrates superior object segmentation performance across all datasets, significantly outperforming baselines in the ARI-O metric. The visualization results further illustrate that our object segmentation masks are finer and more accurate, with minimal background inclusion. Compared to SIMONe and OCLOC, which use low-capacity mixture decoders, models employing high-capacity diffusion decoders demonstrate notable improvements. Additionally, training with multiple viewpoints allows for more comprehensive object representations, which effectively addresses challenges like occlusion in single-viewpoint. Notably, our proposed active viewpoint selection strategy outperforms random selection using the same model architecture. By active selecting the unknown viewpoint with the highest information gain as the next observation, our model enhances the accuracy of viewpoint-independent object representations. On the CLEVRTEX dataset, however, the ARI-A score is lower due to the complexity of lighting and textures in the background, which can lead to the background being divided into two parts. Despite this, the overall segmentation quality, as indicated by the mIoU, remains superior to that of the other methods.

Table 1: **Comparison of segmentation, reconstruction, and generation results.** All values represent the mean of three trials. The best scores are in bold, and the second-best scores are underlined.

| Dataset | Method | ARI-A ↑ | ARI-O ↑ | mIoU ↑ | LPIPS ↓ | FID ↓ |
|---|---|---|---|---|---|---|
| CLEVRTEX | SIMONe | 8.0 | 24.0 | 15.3 | 0.636 | 338.1 |
| | OCLOC | **56.9** | 75.8 | 44.1 | 0.497 | 173.5 |
| | LSD | 52.9 | 78.1 | 45.7 | **0.153** | 110.9 |
| | Ours (Random) | 23.9 | 84.9 | 52.7 | 0.178 | 89.1 |
| | Ours (Active) | 24.3 | **86.1** | **54.3** | 0.175 | **80.3** |
| GSO | SIMONe | 41.5 | 45.8 | 47.2 | 0.481 | 276.8 |
| | OCLOC | 59.5 | 69.8 | 48.6 | 0.431 | 178.0 |
| | LSD | 35.7 | 72.4 | 43.7 | **0.162** | 98.9 |
| | Ours (Random) | 65.3 | 79.6 | 62.3 | 0.176 | 96.5 |
| | Ours (Active) | **68.9** | **82.2** | **64.4** | 0.172 | **87.2** |
| ShapeNet | SIMONe | 32.5 | 40.9 | 41.7 | 0.544 | 325.8 |
| | OCLOC | 49.9 | 69.0 | 42.5 | 0.479 | 239.2 |
| | LSD | 52.0 | 71.1 | 48.5 | **0.172** | 106.3 |
| | Ours (Random) | 54.7 | 71.6 | 53.5 | 0.183 | 103.7 |
| | Ours (Active) | **58.0** | **75.2** | **58.6** | 0.175 | **99.2** |

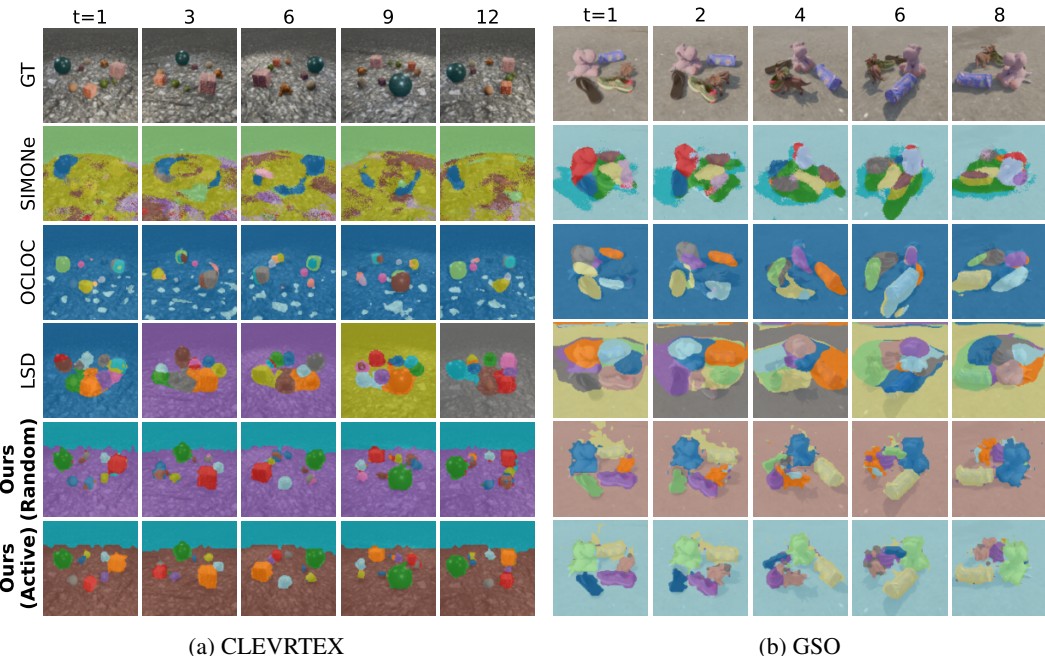

(a) CLEVRTEX        (b) GSO

Figure 3: Visualization of segmentation results on CLEVRTEX and GSO.

## 4.2 Scene Reconstruction

We present quantitative experimental results in Table 1 and visualize qualitative scene reconstruction results in Figure 4. As shown in the results, mixture-based decoder methods SIMONe and OCLOC produce blurry images with a lower LPIPS score, which reflects less perceptually accurate reconstructions. In contrast, diffusion-based decoder methods, including LSD and our model, more effectively capture fine-grained textures and scene details, resulting in clearer, more realistic images. Our model achieves the second-best LPIPS score across all datasets, with LSD performing slightly better. This is because LSD utilizes viewpoint-specific slots to reconstruct images from each viewpoint, allowing for more precise optimization of slots for different viewpoints. In contrast, our model employs

viewpoint-independent slots to reconstruct all viewpoints, requiring more accurate representations to maintain consistent, high-quality reconstructions across multiple viewpoints.

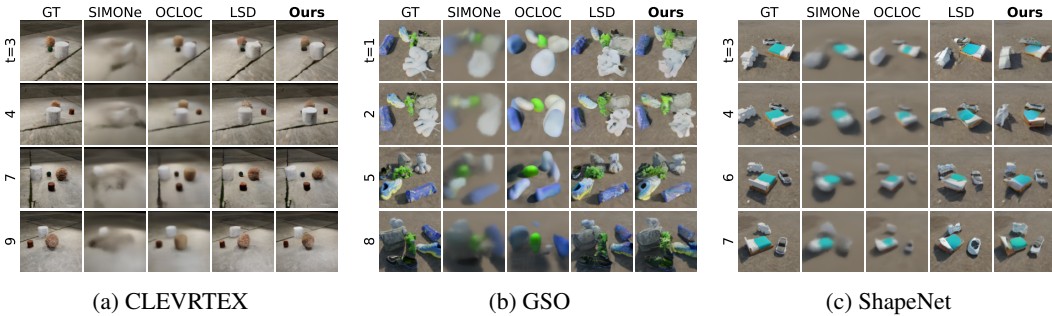

(a) CLEVRTEX         (b) GSO         (c) ShapeNet

Figure 4: Visualization of reconstruction results on CLEVRTEX, GSO, and ShapeNet.

## 4.3 Compositional Generation

To evaluate generation quality, we measure FID across all methods, following SlotDiffusion [12]. Specifically, we first infer all slots from the test set, then randomly shuffle and regroup slots to create a new test set. Finally, samples are generated using either the mixture-based decoder or the diffusion decoder. As shown by the quantitative results in Table 1, our method achieves the lowest FID score, highlighting its superior generation quality.

Our model can generate compositional scene images not only from a single viewpoint, as LSD [11] does, but also across multiple viewpoints. It can sample and interpolate viewpoint annotations to create images from various viewpoints. As shown in Figure 5(a), we use timesteps as viewpoint annotations to generate images of the same scene from 8 different viewpoints. Furthermore, as demonstrated in Figure 5(b), we can extend these 8 timesteps to 12 through interpolation, with the mapped timesteps given by $t' = \frac{8+1}{12+1} t$ ($t = [1, ..., 8]$).

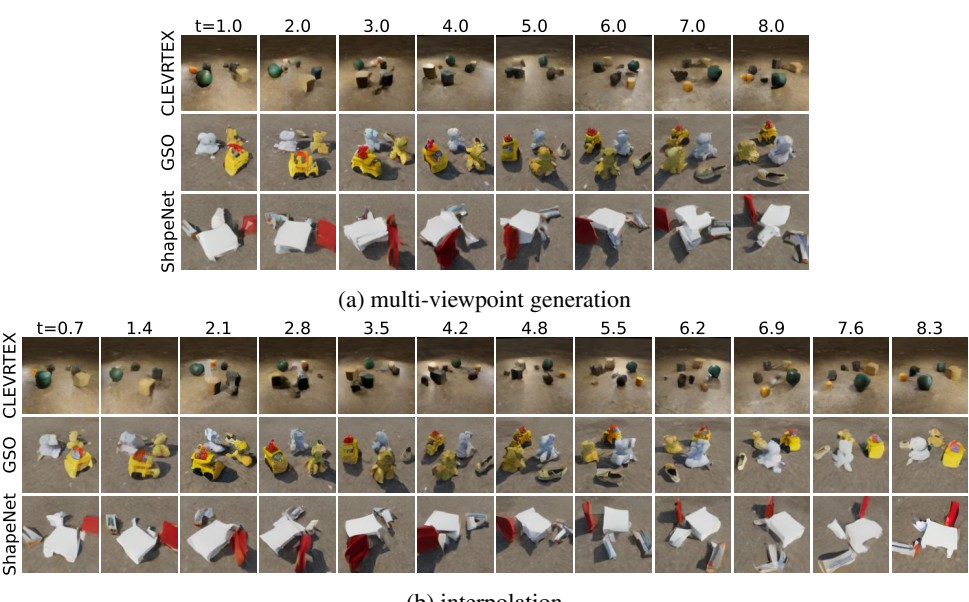

(a) multi-viewpoint generation

(b) interpolation

Figure 5: Multi-viewpoint compositional generation samples and interpolation.

## 4.4 Novel Viewpoint Synthesis

Novel viewpoint synthesis is a key feature of our model, achieved through the active viewpoint selection strategy that predicts images from unknown viewpoints ($\mathcal{P}$) based on viewpoint-independent

object-centric representations from known viewpoints images ($\mathcal{O}$) and the viewpoint timesteps corresponding to the target viewpoint. This synthesis process aligns with the steps in Algorithm 2. First, the active viewpoint selection strategy is used to obtain optimal slots $\mathcal{S}$. Then, $\mathcal{S}$ is combined with viewpoint representations $\mathcal{S}^{\text{view}}$ from $\mathcal{P}$, which serve as conditions for the slot-conditioned diffusion decoder to generate images for the target viewpoints. Notably, the diffusion decoder produces these images without additional masks. Therefore, segmentation masks for the novel images are inferred using Algorithm 1, with the generated image $\tilde{x}$ from $\mathcal{P}$ as the input. Figure 6 presents the results, where black viewpoint timesteps indicate actively selected viewpoints and red viewpoint timesteps represent predicted viewpoints. It can be seen that our model accurately synthesizes images from novel viewpoints and obtains corresponding segmentation results.

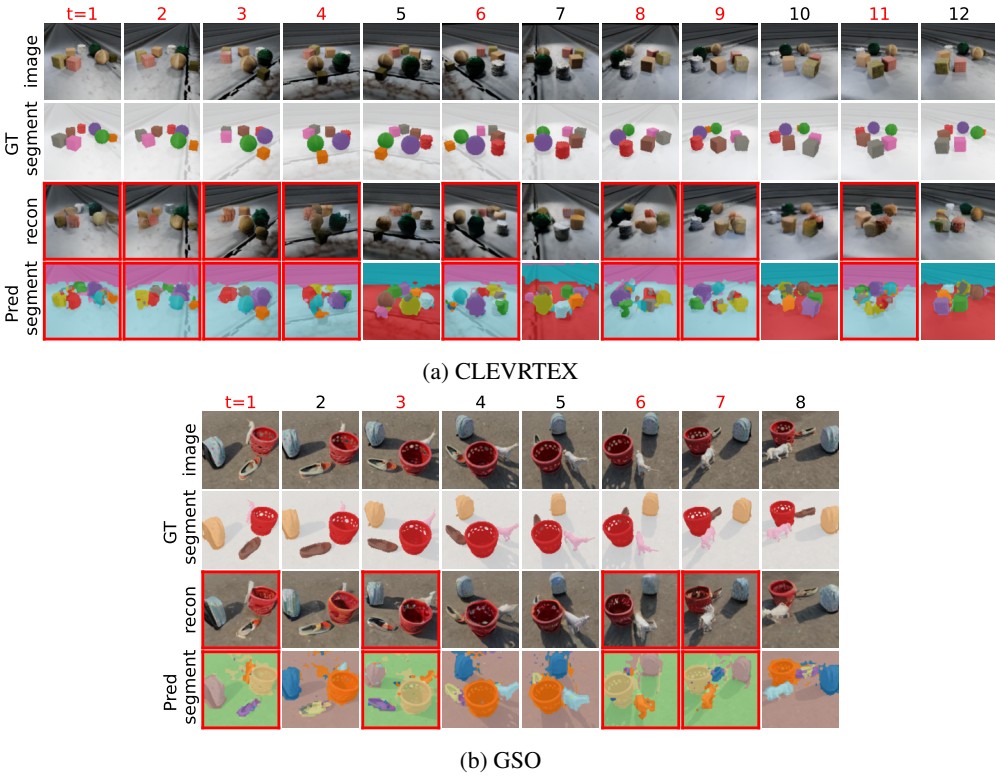

Figure 6: Novel viewpoint synthesis results on CLEVRTEX and GSO.

## 5 Conclusion

We propose AVS, a multi-viewpoint object-centric learning model featuring an active viewpoint selection strategy. AVS exhibits outstanding performance in unsupervised object segmentation and image generation. Compared to the random viewpoint selection strategy, our active viewpoint selection strategy significantly improves viewpoint-independent object-centric representations, enabling the model to better understand and perceive visual scenes. Moreover, our model can predict images from unknown viewpoints and generate images with novel viewpoints.

**Limitation** Although our model performs well on the dataset presented in this article, the active selection process exhibits high training complexity, which is positively correlated with the number of selected viewpoints and the diffusion sampling steps. While reducing the number of diffusion sampling steps can improve efficiency, the computational demands during training remain significant. One potential improvement is to use reconstructed features instead of reconstructed images, thereby downscaling from a high-dimensional image space to a low-dimensional vector space. This approach can significantly reduce the computational complexity associated with predicting unknown viewpoints. Another solution is to implement a more direct viewpoint selection mechanism that outputs the next observation viewpoint directly, rather than predicting each unknown viewpoint through traversal.

## Acknowledgments and Disclosure of Funding

This work was supported in part by the National Natural Science Foundation of China No. 62176060, STCSM project No. 22511105000, Lenovo Scientists Program (Study on Prior-based Visual Scene Analysis), the Shanghai Platform for Neuromorphic and AI Chip under Grant 17DZ2260900 (NeuHelium), and the Program for Professor of Special Appointment (Eastern Scholar) at Shanghai Institutions of Higher Learning.

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

# A    Details of Datasets

The dataset configuration used in this paper is provided in Table 2. We modified the official CLEVRTEX [25] and Kubric [28] codebases to generate the multi-viewpoint dataset. Specifically, spherical coordinates are used to assign the camera position $(x, y, z)$ for each viewpoint in the scene. The camera coordinates as a function of spherical coordinates are given by:

$$
\begin{aligned}
x &= \rho \sin\theta \cos\phi \\
y &= \rho \sin\theta \sin\phi \\
z &= \rho \cos\theta
\end{aligned}
\tag{5}
$$

where $\rho$ is the radius distance from the origin, $\theta$ is the elevation angle with respect to the positive $z$-axis, and $\phi$ is the azimuth angle in the $xy$-plane measured from the positive $x$-axis.

Table 2: Configurations of datasets

| Datasets | CLEVRTEX | | | GSO/ShapeNet | | |
|---|---|---|---|---|---|---|
| Split | Train | Valid | Test | Train | Vaid | Test |
| # of Images | 5000 | 100 | 100 | 5000 | 100 | 100 |
| # of Objects | 3∼10 | | | 3∼6 | | |
| # of Views | 12 | | | 8 | | |
| Image Size | 128×128 | | | | | |
| Distance $\rho$ | [10.5,12] | | | | | |
| Elevation $\theta$ | [0.15π,0.3π] | | | | | |
| Azimuth $\phi$ | [0,2π] | | | | | |

# B    Choices of Hyperparameters

**SIMONe**    We implement SIMONe using the PyTorch framework. The architecture and hyperparameters used to train SIMONe closely followed the original paper except 1) the number of slots was 12 for CLEVRTEX and 8 for GSO and ShapeNet; 2) the local batch size was 2.

**OCLOC**    The official OCLOC implementation[2] was used. The model for CLEVRTEX was trained with the default hyperparameters described in the "exp_multi/model/blender.yaml" file of the official code repository, except for the number of slots, which was set to 12. The models for GSO and ShapeNet were trained with the default hyperparameters described in the "exp_multi/model/kubric.yaml" file of the official code repository.

**LSD**    The official LSD implementation[3] was used. The hyperparameters for CLEVRTEX were similar to the ones described in the original LSD paper for CLEVRTEX, except 1) the input resolution was 128; 2) the local batch size was 128. The hyperparameters for GSO and ShapeNet were similar to the ones described in the original LSD paper for MOVi-C, except 1) the input resolution was 128; 2) the number of slots was 8.

**Ours**    The architecture and hyperparameters of our model are presented in Table 3. Following DINOSAUR [32], we utilize the pretrained DINO to extract features from images and reconstruct these features from object representations using a MLP Decoder. Additionally, we employ a Viewpoint Encoder module to obtain viewpoint representations, which consists of a MLP with linear layers and ReLU activation functions. The hyperparameter $\lambda$ in Eq.(4) is set to 0 for the first 20k training steps. After this period, it increases linearly with the number of training steps, reaching 0.5 at the maximum training step.

# C    Computation Requirement

We compare the computational requirements of our model with baseline models in the CLEVRTEX setting. Our model requires approximately 22 GB of training memory per GPU, compared to 15 GB

---

[2] https://github.com/jinyangyuan/multiple-unspecified-viewpoints
[3] https://github.com/JindongJiang/latent-slot-diffusion

Table 3: Hyperparameters of our model used in experiments.

| Module | Hyperparameter | CLEVRTEX | GSO | ShapeNet |
|---|---|---|---|---|
| **General** | Batch Size | 32 | 24 | 16 |
| | Training Steps | | 200K | |
| | # Viewpoints | | 4 | |
| **DINO** | Input Resolution | | 224 | |
| | Patch Size | | 8 | |
| | # Patches | | 28×28 | |
| | Output Channels | | 384 | |
| **Viewpoint Encoder** | Input Channels | 2 | 5 | 5 |
| | Channel Multipliers | | [512, 512] | |
| | Output Channels | 16 | 32 | 32 |
| | Learning Rate | 1e-4 | 3e-5 | 3e-5 |
| **Slot Attention** | Input Resolution | | 28 | |
| | Input Channels | | 384 | |
| | # Iterations | | 3 | |
| | Slot Attr Size | 64 | 128 | 128 |
| | Slot View Size | 16 | 32 | 32 |
| | # Slots | 11 | 8 | 8 |
| | Learning Rate | 1e-4 | 3e-5 | 3e-5 |
| **Auto-Encoder** | Model | | KL-8 | |
| | Input Resolution | | 128 | |
| | Output Resolution | | 16 | |
| | Output Channels | | 4 | |
| **MLP Decoder** | Input Channels | 144 | 160 | 160 |
| | Channel Multipliers | | [1024, 1024, 1024] | |
| | Output Channels | | 384 | |
| | Output Resolution | | 28 | |
| **LSD Decoder** | Input Resolution | | 16 | |
| | Input Channels | | 4 | |
| | $\beta$ scheduler | | Linear | |
| | Mid Layer Attention | | Yes | |
| | # Res Blocks / Layers | | 2 | |
| | Image Latent Scaling | | 0.18215 | |
| | Learning Rate | 1e-4 | 3e-5 | 3e-5 |
| | # Heads | 4 | 4 | 8 |
| | Base Channels | 144 | 160 | 160 |
| | Attention Resolution | [1, 2, 4] | [1, 2, 4, 4] | [1, 2, 4, 4] |
| | Channel Multipliers | [1, 2, 4] | [1, 2, 4, 4] | [1, 2, 4, 4] |

for SIMONe, 23 GB for OCLOC, and 21 GB for LSD. We train our model on 4 NVIDIA RTX 4090 GPUs over 4.5 days, while SIMONe is trained in 1.5 days, OCLOC in 2.5 days, and LSD in 1.5 days, all using the same GPU setup.

## D   Extra Experimental Results

In this section, we provide additional experimental results to demonstrate our model's capabilities.

Table 4 presents the complete segmentation and reconstruction results, including the standard deviation values from three tests. The proposed method outperforms the compared methods in most cases.

Table 4: **Full table of segmentation and reconstruction performance.** Extended results from the main text, now including standard deviation values.

| Dataset | Method | ARI-A ↑ | ARI-O ↑ | mIoU ↑ | LPIPS ↓ |
|---|---|---|---|---|---|
| CLEVRTEX | SIMONe | 8.0±0.01 | 24.0±0.02 | 15.3±0.01 | 0.636±5e-5 |
| | OCLOC | **56.9±0.3** | 75.8±0.2 | 44.1±0.3 | 0.497±1e-3 |
| | LSD | 52.9±0.05 | 78.1±0.08 | 45.7±0.02 | **0.153±3e-4** |
| | Ours (Random) | 23.9±0.04 | 84.9±0.06 | 52.7±0.3 | 0.178±3e-4 |
| | Ours (Active) | 24.3±0.01 | **86.1±0.08** | **54.3±0.09** | 0.175±8e-4 |
| GSO | SIMONe | 41.5±0.02 | 45.8±0.01 | 47.2±0.04 | 0.481±1e-4 |
| | OCLOC | 59.5±0.01 | 69.8±0.2 | 48.6±0.02 | 0.431±9e-4 |
| | LSD | 35.7±0.05 | 72.4±0.1 | 43.7±0.07 | **0.162±4e-4** |
| | Ours (Random) | 65.3±0.03 | 79.6±0.2 | 62.3±0.3 | 0.176±7e-4 |
| | Ours (Active) | **68.9±0.02** | **82.2±0.1** | **64.4±0.3** | 0.172±6e-4 |
| ShapeNet | SIMONe | 32.5±0.03 | 40.9±0.06 | 41.7±0.06 | 0.544±1e-4 |
| | OCLOC | 49.9±0.2 | 69.0±0.2 | 42.5±0.4 | 0.479±5e-4 |
| | LSD | 52.0±0.08 | 71.1±0.02 | 48.5±0.1 | **0.172±2e-4** |
| | Ours (Random) | 54.7±0.05 | 71.6±0.09 | 53.5±0.1 | 0.183±1e-3 |
| | Ours (Active) | **58.0±0.03** | **75.2±0.07** | **58.6±0.08** | 0.175±9e-4 |

## D.1 Multi-Viewpoint Image Editing

Since our model learns viewpoint-independent object-centric representations from multi-viewpoint images, it can perform image editing consistently across different viewpoints. As shown in Figure 7, our model enables manipulation of objects across scenes, including removal, insertion, and swapping. When removing an object, the previously obscured areas of other objects are automatically filled in, with even the shadows of objects accurately restored. When inserting an object from one scene into another, images from different viewpoints consistently display the new object from the appropriate viewpoints. Additionally, our model can swap objects between two scenes.

## D.2 Additional Visualizations

In Figure 8-13, we visualize additional qualitative results of our model across different datasets.

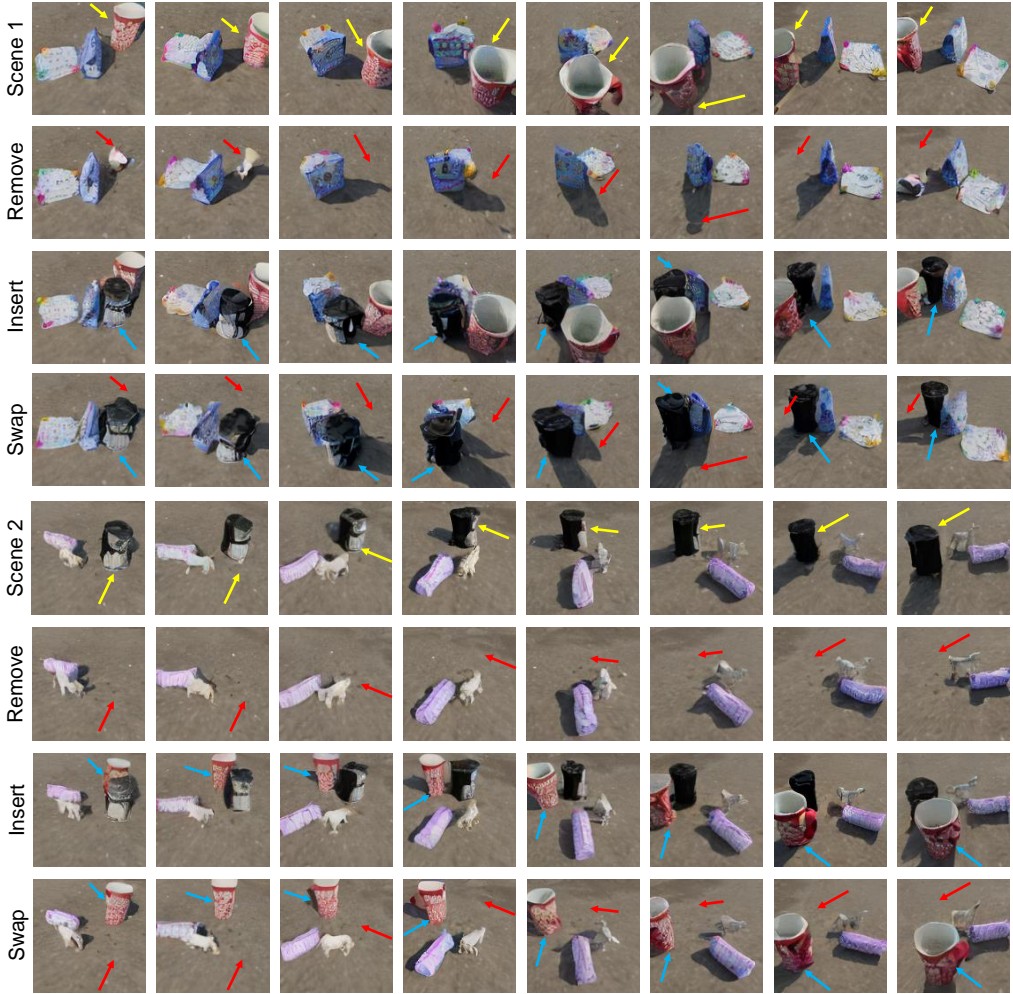

Figure 7: **Multi-Viewpoint image editing.** Our model enables object manipulation across different viewpoints, including object removal, insertion, and swapping. In the figure, yellow arrows indicate objects randomly selected for manipulation, red arrows indicate objects removed from the scene, and blue arrows point to newly inserted objects from another scene.

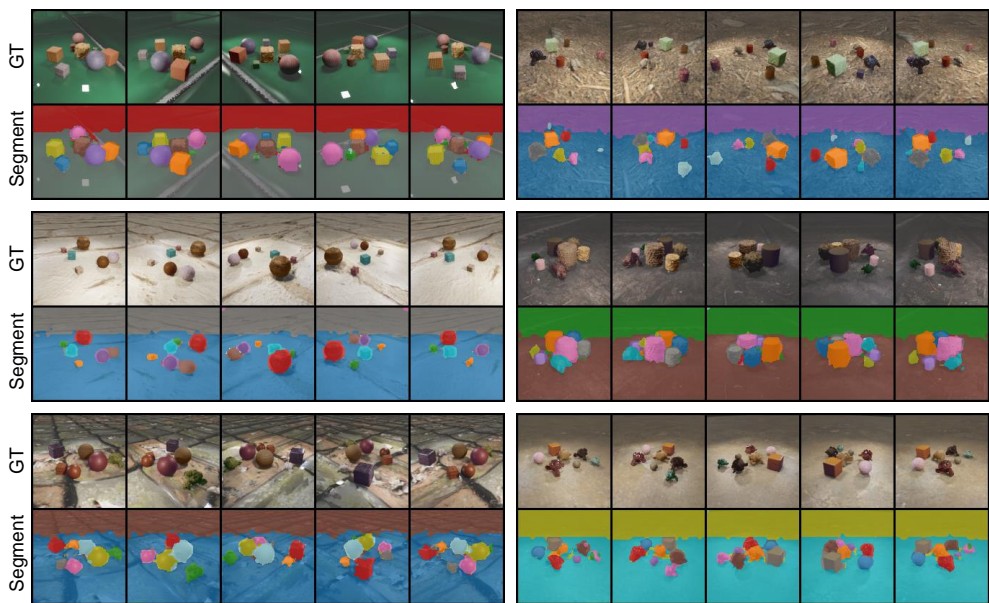

Figure 8: Unsupervised object segmentation results on CLEVRTEX.

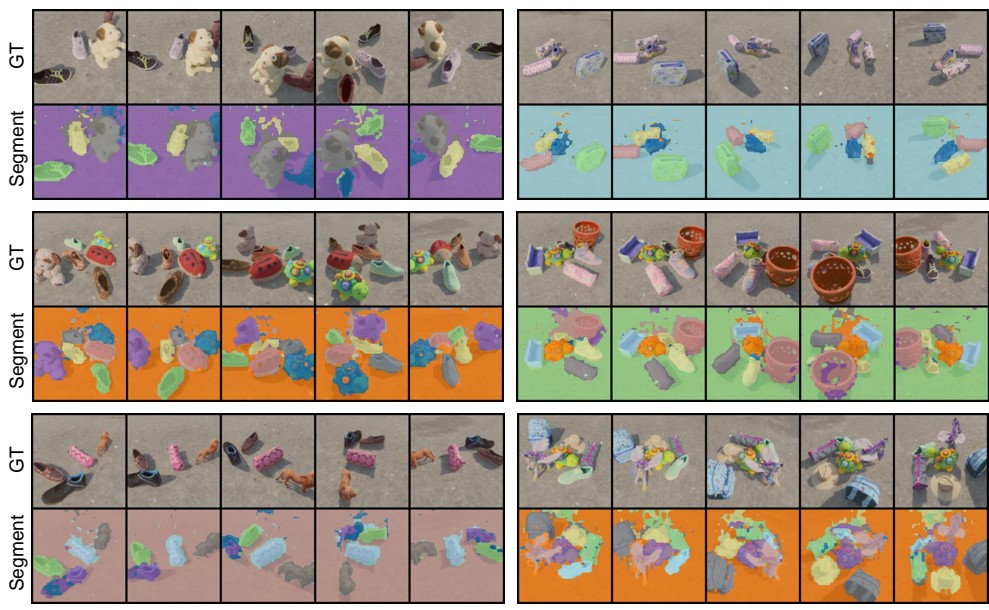

Figure 9: Unsupervised object segmentation results on GSO.

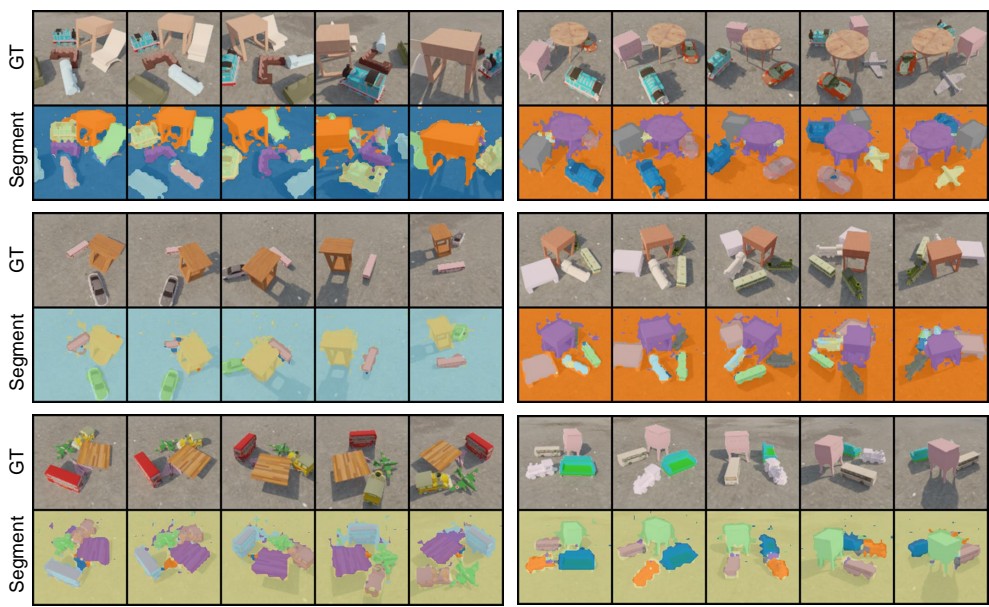

Figure 10: Unsupervised object segmentation results on Shapenet.

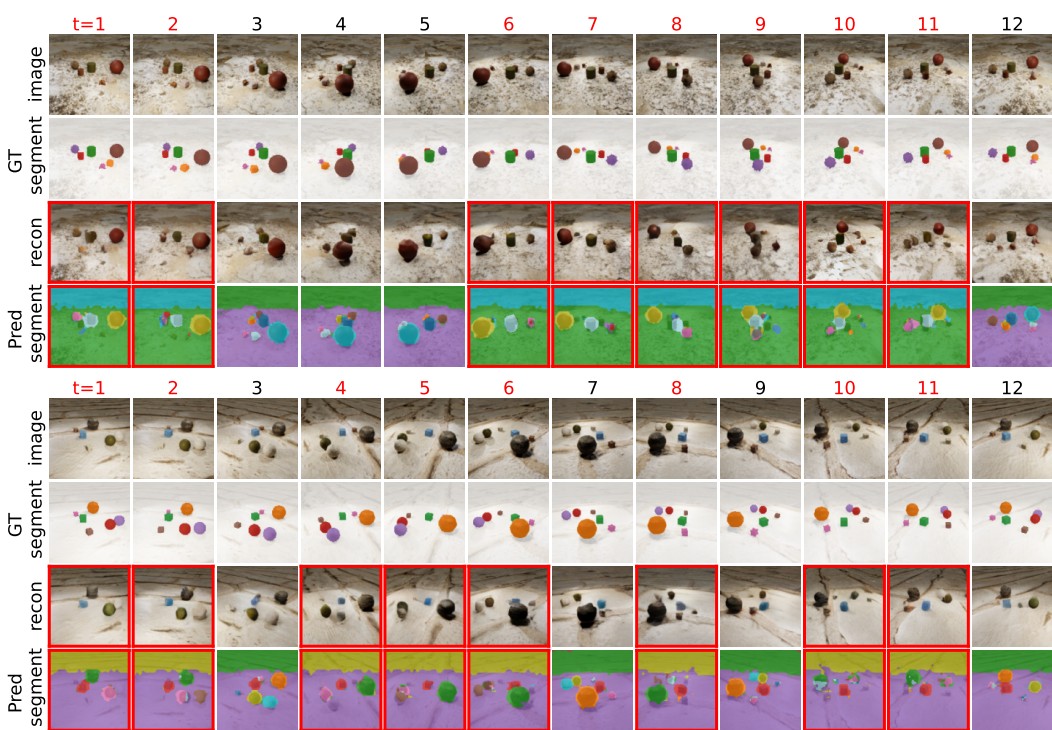

Figure 11: Novel viewpoint synthesis results on CLEVRTEX.

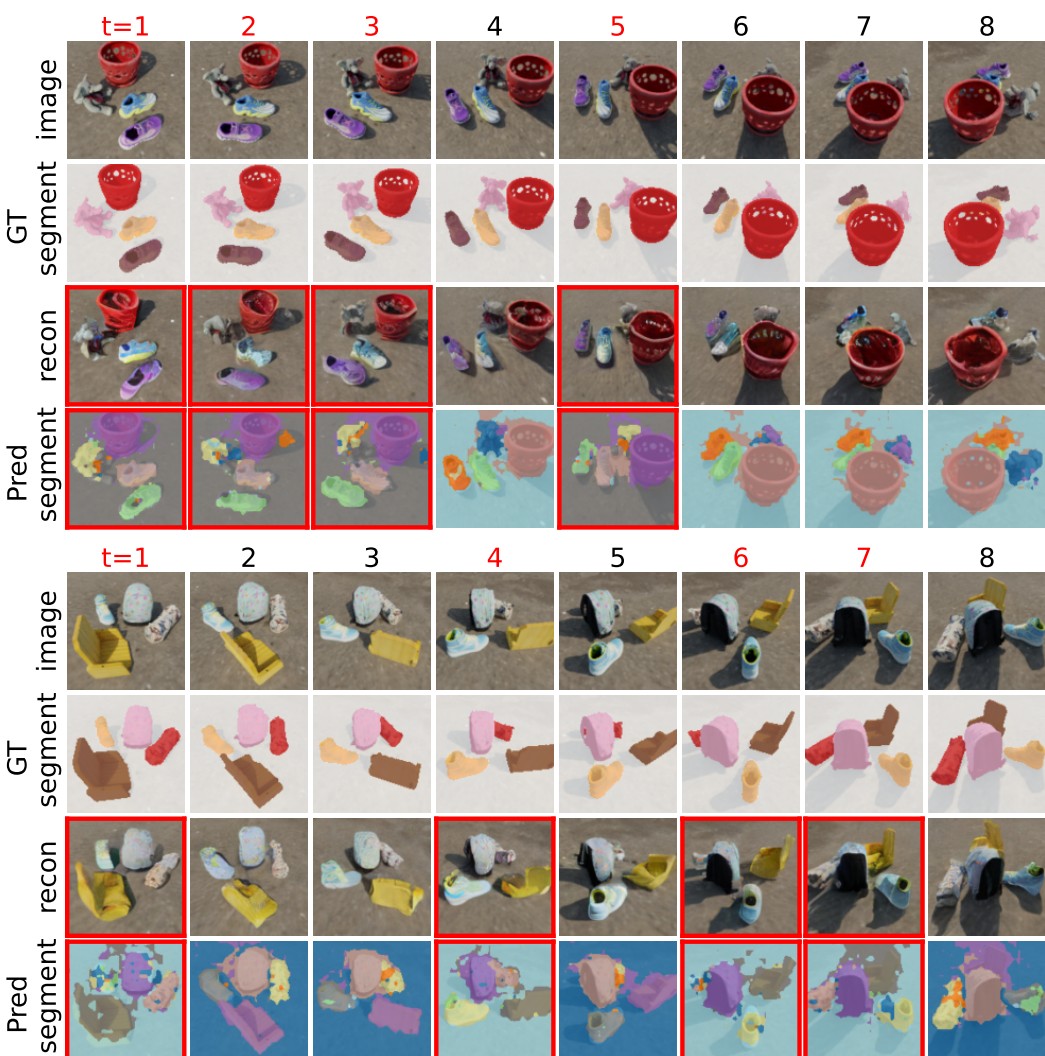

Figure 12: Novel viewpoint synthesis results on GSO.

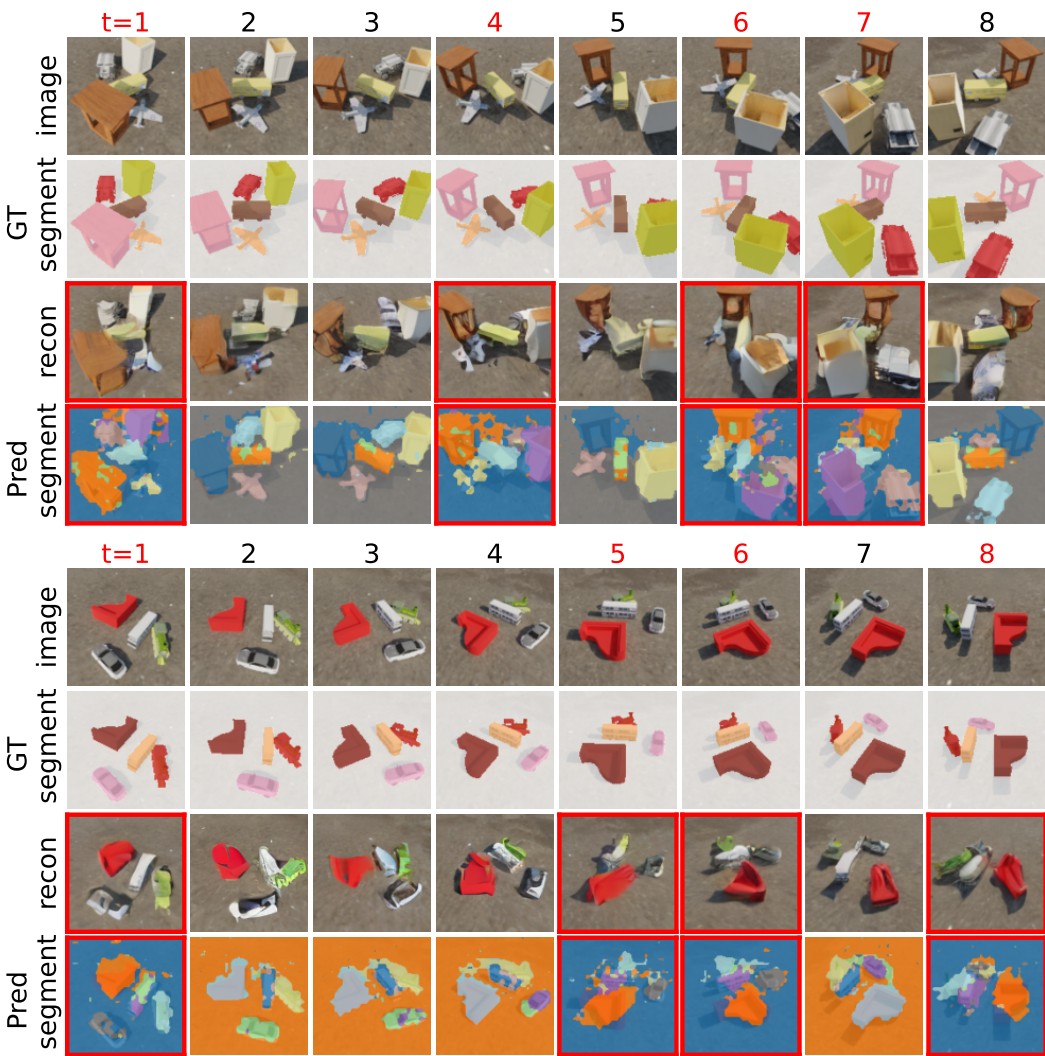

Figure 13: Novel viewpoint synthesis results on ShapeNet.

