# OpenReview forum: "Improving Viewpoint-Independent Object-Centric Representations through Active Viewpoint Selection"
_NeurIPS.cc/2024/Conference — NeurIPS 2024 poster_

### Official Review · Reviewer_qk67 · 2024-07-12

**Soundness:** 2
**Presentation:** 3
**Contribution:** 3
**Rating:** 5
**Confidence:** 3

**Summary:**

This paper introduces an image selection method designed to incrementally enrich an observation set with informative images from an extensive unknown set, aiming to maximize information gain with a limited or specific number of images. The methodology innovatively integrates Multi-Viewpoint Slot Attention for getting object-centric representations, enabling the gradual identification of the most informative images through a comparative analysis of representations before and after the inclusion of newly generated images. Additionally, it leverages a diffusion model in conjunction with slot attention and viewpoint encoding to synthesize images for image prediction and new viewpoint validation. The experimental outcomes demonstrate improvements in segmentation and reconstruction tasks, underscoring the effectiveness of the proposed approach.

**Strengths:**

+The paper presents a novel method that effectively disentangles content and viewpoint information using Multi-Viewpoint Slot Attention. This decoupling approach could offer a fresh perspective in the field.

+The validation of the proposed method across four task scenarios provides a comprehensive evaluation of its advantages in supporting various downstream multi-view applications.

**Weaknesses:**

-This method may be time consuming, since there are two loops in the selection strategy. It is recommended to conduct a detailed analysis of the trade-off between processing time and performance gains, potentially including a comparative study with alternative methods to gain further insights into the method's efficiency.

-While the method claims the ability to accurately predict images from unknown viewpoints, the exact benefit of the viewpoint selection strategy in this context remains unclear. Moreover, Can the S_view be obtained in image prediction tasks?

-The necessity of the generating Prediction Set within the proposed framework may not be very clear. Based on the pipeline depicted in Figure 2, an investigation into the framework's performance when the Prediction Set is set as same as Unknown Set could provide valuable insights into the role and impact of the Prediction Set. A comparative experiment exploring this scenario is suggested.

**Questions:**

Please refer to the Weaknesses

**Limitations:**

Limitation is discussed in the work and the work will not have any potential negative societal impact.

---

> ### Author Rebuttal · Authors · 2024-08-07
>
> We sincerely thank you for your valuable feedback and constructive comments. We have carefully considered each point raised and provide our responses below.
>
> **1. Computational Complexity**
>
> We acknowledge that our method may be more time-consuming due to the presence of two loops in the selection strategy. However, it is important to highlight that this increased computational complexity is offset by our method's unique capability to predict unknown viewpoints, a feature not available in alternative methods. Additionally, our approach demonstrates superior performance in scene segmentation compared to other methods.
>
> **2. Benefit of the Viewpoint Selection Strategy**
>
> We appreciate your observation regarding the role of the viewpoint selection strategy in predicting images from unknown viewpoints.
>
> - The viewpoint selection strategy is integral to enhancing the quality of viewpoint-independent object-centric representations. It ensures that the most informative viewpoints are selected for observation, thereby reducing redundancy and improving prediction accuracy. Accurately predicting images from unknown viewpoints is both a necessary condition for the effectiveness of our viewpoint selection strategy.
> - $S^{view}$ can be obtained in image prediction tasks by inputting the target viewpoint timestep into the viewpoint encoder. The model uses $S^{view}$ to guide the prediction of the image from target viewpoint.
>
> **3. Necessity of the Prediction Set**
>
> We apologize if Figure 2 is not sufficiently clear. In our framework, the images from the Unknown Set are not directly accessible to the model. Only the viewpoint timesteps of the Unknown Set are available. The Prediction Set consists of images predicted from these viewpoint timesteps. Generating the Prediction Set is a necessary step to estimate the information increment of the unknown viewpoints, which plays a crucial role in our viewpoint selection strategy.

---

### Official Review · Reviewer_gp25 · 2024-07-12

**Soundness:** 3
**Presentation:** 2
**Contribution:** 3
**Rating:** 5
**Confidence:** 3

**Summary:**

This work introduces a method for multi view object centric reconstruction. The authors extend a previous work LSD[10] from single view to multiple unposed views of the same scene and introduce an active view selection mechanism at training time. The method takes N views, decomposes the scene into K slots and decodes them through a conditioned latent diffusion model. During training an active viewpoint mechanism is used, rather than random sampling sets of viewpoints, as a form of hard mining that helps the model converge to better solutions. The authors evaluate their proposal on synthetically created datasets based on CLEVR-TEX, GSO and ShapeNet where they demonstrate better unsupervised segmentation of objects, scene reconstruction and novel view synthesis on par or better wrt to previous approaches not based on diffusion decoders.

**Strengths:**

+ Strong improvement wrt to previous multi-view methods that did not use a diffusion based decoder.

+ Additional improvements achieved using something resembling hard mining while training that while making training more complex do not have side effect on the complexity at inference time.

+ The paper introduces 3 synthetic multi-view datasets to evaluate the model that could be used by the community to further develop this specific field if the code to replicate them will be made available.

**Weaknesses:**

a. Presentation can be improved. In particular Sec. 3.1.2 would benefit by more details in the text. For example it is not clear to me what $h_m$ or $v$ in the pseudo code are. Also in terms of figures and plots many of the results are really small and hard to see. I would have preferred less results but bigger in the main paper and the rest in the appendix. Also I feel like Tab 5 should be in the main paper in place of Fig. 3 and Tab. 1 and 2.

b. Limited evaluation. While on one hand it is nice that the authors contributed a new set of 3 datasets to evaluate their method, these are rather small and synthetic. UNfortunately this is inline with most of the competitors the authors compare to, but would be nice to see the field move more towards realistic scenarios. An additional note is that is not clear to me why the authors re-rendered datasets rather than using the one [made available by OCLOC](https://huggingface.co/datasets/jinyangyuan/ocloc-data) which per my understanding should be quite similar.

c. Slightly unclear evaluation settings. I have not found in the paper clear mention on how the unsupervised segmentation is obtained from the model (Sec. 4.2) or how the temporal timesteps used in Sec. 4.3 and 4.4 are injected into the model to make it conditioned on a certain viewpoint. Are they additional conditioning input to the latent diffusion model?

**Questions:**

**Questions**


1. Do you have an intuition on  the big discrepancy in the ranking defined by LPIPS and FID in Tab. 1 and 2? Usually the two metrics tend to be very correlated in my experience.

2.  Can you clarify my doubts with respect to weakness [b] and weakness [c]?

**Typos**

* Discrepancy between Fig. 3 and Tab. 5 → ARI-O == ARI-FG?

**Limitations:**

Some limitations have been discussed in the manuscript.

---

> ### Author Rebuttal · Authors · 2024-08-07
>
> We sincerely thank you for your valuable feedback and constructive comments. We have carefully considered each point raised and provide our responses below.
>
> **1. Presentation**
>
> Thank you for your feedback on the presentation. We will revise Section 3.1.2 to provide more details and clarity regarding the notation used in the pseudocode. These variables are defined as follows: $h_m$ represents the image features extracted by DINO, and $v$ represents the value function of the attention mechanism. We agree that larger figures and plots would enhance readability, and will enlarge key figures and plots in the main paper, moving additional results to the appendix to maintain focus.
>
> **2. Dataset Selection and Re-Rendering**
>
> - Our choice to re-render datasets rather than using the OCLOC datasets was primarily due to the need for specific viewpoint parameters ($\rho$, $\phi$, $\theta$) that were not available in the OCLOC datasets. These parameters were initially intended to be used in our method, but we later shifted to using timesteps for simplicity.
> - For the datasets, we aimed to maintain consistency with the configuration and rendering methods used in OCLOC. However, we chose the CLEVRTEX dataset over the simpler CLEVR and SHOP datasets used by OCLOC to provide more complex scenes with richer textures and variability, which better test the model's capabilities.
>
> **3. Real-World Data Evaluation**
>
> We understand the importance of assessing the model's generalization to real-world data. However, there is currently a lack of suitable multi-viewpoint real-world datasets for comprehensive evaluation of our model and similar methods. This poses a challenge in demonstrating real-world applicability. Nonetheless, the objects in the GSO and ShapeNet datasets closely resemble real-world objects, providing a realistic basis for testing our model's generalization beyond purely synthetic environments. These datasets effectively bridge the gap between synthetic and real-world scenes.
>
> **4. Evaluation Settings**
>
> We appreciate the reviewer's feedback regarding the clarity of the evaluation settings. We apologize for any confusion and provide the following clarifications:
>
> - The unsupervised segmentation results are derived from the attention masks computed within the multi-viewpoint slot attention module.
> - The temporal timesteps are encoded using the viewpoint encoder. These encoded timesteps are then concatenated with the viewpoint-independent object-centric representations to serve as conditioning input for the diffusion model, guiding the image generation process.
>
> **5. Discrepancy Between LPIPS and FID Rankings**
>
> Thank you for pointing out the discrepancy between LPIPS and FID rankings in Tables 1 and 2. This discrepancy may arise from the different aspects of image quality they measure. LPIPS evaluates perceptual similarity at a fine detail level, while FID assesses global distributional similarity. The models being evaluated might perform differently in preserving local details versus maintaining overall image realism and distribution, leading to variations in how each metric ranks the results.
>
> **6. Discrepancy Between Fig. 3 and Tab. 5 → ARI-O == ARI-FG?**
>
> Thank you for pointing out this discrepancy. We apologize for any confusion caused. The notation ARI-O and ARI-FG should indeed be consistent across Figure 3 and Table 5.

---

> > ### Comment · Reviewer_gp25 · 2024-08-12
> > **Acknowledgement**
> >
> > Thanks authors for providing a rebuttal and addressing my questions.
> >
> > I would suggest to include these discussions in the next version fo the manuscript.
> >
> > I tend to disagree with this strong claim made in the rebuttal
> >
> > "Nonetheless, the objects in the GSO and ShapeNet datasets closely resemble real-world objects, providing a realistic basis for testing our model's generalization beyond purely synthetic environments. **These datasets effectively bridge the gap between synthetic and real-world scenes**."
> >
> > Such strong claims would require very substantial evidences. From my experience If this was true the domain adaptation litterature would not exhist in the first place.

---

> > > ### Author Response · Authors · 2024-08-14
> > >
> > > Thank you for your feedback and for taking the time to review our rebuttal. We appreciate your suggestion to include these discussions in the next version of the manuscript, and we will certainly consider that in our revisions.
> > >
> > > Regarding the statement about the GSO and ShapeNet datasets, we acknowledge your concerns and agree that the claim may have been overstated. Our intent was to emphasize that while these datasets are synthetic, the objects they contain closely resemble real-world objects. However, we fully recognize that there remains a significant gap between synthetic datasets and real-world environments.
> > >
> > > We will revise our statement to avoid making an overly strong claim. We appreciate your guidance in ensuring that our manuscript accurately conveys the limitations and strengths of our approach.

---

### Official Review · Reviewer_qAwd · 2024-07-14

**Soundness:** 2
**Presentation:** 2
**Contribution:** 2
**Rating:** 3
**Confidence:** 3

**Summary:**

This paper describes a novel active viewpoint selection strategy (AVS) for enhancing multi-viewpoint object-centric learning methods. The core idea is to select the most informative viewpoints actively rather than using random or sequential strategies, which can be inefficient and may omit critical scene information. The model enhances viewpoint-independent object-centric representations, leading to better understanding and perception of visual scenes. It can predict and generate images from unknown viewpoints.

**Strengths:**

1. The active viewpoint selection strategy (AVS) is a novel concept that addresses the limitations of traditional random or sequential viewpoint selection methods in multi-viewpoint object-centric learning. The paper demonstrates through experiments that AVS significantly enhances the performance of segmentation and reconstruction tasks compared to random viewpoint selection strategies.
2. The proposed method leads to better viewpoint-independent object-centric representations, which are crucial for accurately understanding and perceiving visual scenes from various angles.
3. The model's ability to predict images from unknown viewpoints is a significant strength, allowing it to work effectively even with limited observational data.  The model can generate images with novel viewpoints that it hasn't been explicitly trained on, showcasing its generative capabilities and the robustness of the learned representations.
4. Despite using fewer viewpoints for training, the proposed model achieves superior results, indicating that it can efficiently learn comprehensive representations.
5. Evaluation: The paper includes a thorough evaluation using multiple datasets and various metrics, providing a comprehensive understanding of the model's strengths and areas of improvement. The model's performance is benchmarked against other contemporary methods, such as SIMONe, OCLOC, and LSD, showing its competitive edge in the field.

**Weaknesses:**

1. The active selection process of the proposed model has high computational complexity, which is directly proportional to the number of selected viewpoints and the diffusion sampling steps. This could affect the training speed and efficiency.
2. Reliance on Viewpoint Continuity: The method's effectiveness is contingent on the continuity of multiple viewpoints. If the viewpoints are not continuous or related, the model may struggle to perform novel view synthesis.
3. Generalization: While the model performs well on the datasets presented in the paper, its generalization capabilities to other datasets or real-world scenarios are not fully explored. The active viewpoint selection strategy assumes that specific scenes may be more sensitive to information from certain viewpoints. This assumption might not hold true for all types of scenes or objects. Although the active selection strategy aims to avoid redundancy or omission of scene information, there is still a possibility that the selected viewpoints may not always capture the most informative aspects of a scene.
4. Overfitting to Synthetic Data: The experiments were conducted on synthetic datasets, which might not fully represent the complexity and variability of real-world data. There is a risk that the model could overfit to the synthetic data and not perform as well on real images.
5. No Discussion on Computational Resources: While the paper mentions the GPU type used, it does not provide a detailed analysis of the computational resources needed for training and inference, which is important for assessing the scalability of the approach.
6. No Open Access to Code and Data: At the time of submission, the paper did not provide open access to the code and data, which is important for reproducibility and further research by the community.

**Questions:**

1. How does the model generalize to real-world datasets and scenes that may have more variability and complexity than the synthetic datasets used in the experiments?
2. Can you provide more details on the computational complexity of the active viewpoint selection process and how it scales with the number of viewpoints and scene complexity?
3. What strategies are employed to make the training process more efficient, given the high training complexity mentioned as a limitation?
4. How does the model perform when the continuity assumption of viewpoints is violated? Are there any fallback mechanisms or alternative strategies?
5. How robust is the model to noise, occlusions, and other common challenges present in real-world visual data?
6. Are there plans to release the code and data used in the experiments to ensure reproducibility and facilitate further research by the community?
7. Can you provide examples of failure cases where the model did not perform well, and what insights can be gained from these cases?
Theoretical Foundation Question:1

**Limitations:**

1. high training cost
2. relying on viewing continunity which might not always hold in real-world scenarios
3.

---

> ### Author Rebuttal · Authors · 2024-08-07
>
> We sincerely thank you for your valuable feedback and constructive comments. We have carefully considered each point raised and provide our responses below.
>
> **1. Computational Complexity**
>
> We acknowledge that our method may be more time-consuming due to the presence of two loops in the selection strategy. However, it is important to highlight that this increased computational complexity is offset by our method's unique capability to predict unknown viewpoints, a feature not available in alternative methods. Additionally, our approach demonstrates superior performance in scene segmentation compared to other methods.
>
> **2. Viewpoint Continuity**
>
> We appreciate your concern regarding the reliance on viewpoint continuity. For scenes involving discontinuous viewpoints, we can use specific viewpoint parameters (denoted as $\rho$, $\phi$, and $\theta$ and defined in Appendix A) instead of relying on viewpoint timesteps. This approach allows for more precise control over the viewpoint and improves the model's ability to handle novel view synthesis.
>
> **3. Generalization and Evaluation on Real-World Data**
>
> We understand the importance of assessing the model's generalization to real-world data. However, there is currently a lack of suitable multi-viewpoint real-world datasets for comprehensive evaluation of our model and similar methods. This poses a challenge in demonstrating real-world applicability. Nonetheless, the objects in the GSO and ShapeNet datasets closely resemble real-world objects, providing a realistic basis for testing our model's generalization beyond purely synthetic environments. These datasets effectively bridge the gap between synthetic and real-world scenes.
>
> **4. Open Access to Code and Data**
>
> We are preparing our code and datasets for public release and plan to make them available on Github upon acceptance of the paper. This will ensure reproducibility and facilitate further research by the community.

---

### Decision · Program_Chairs · 2024-09-25

**Decision:**

Accept (poster)

**Comment:**

This paper received mixed reviews. Reviewer qAwd recommended rejection, citing concerns about computational complexity, the exclusive use of synthetic datasets, and generalization to visual discontinuities. The other reviewers were positive on the paper, but echoed some of these concerns. The authors convincingly rebutted the issues, pointing out that only synthetic datasets are available for this particular problem, and prior publications in NeurIPS follow the same experimental setup. qAwd did not respond to the rebuttal, and there is some concern that he used an LLM to write his review, which is not allowed; hence his review was given less importance than the others. The paper offers interesting novel contributions in using diffusion for generating images within the active viewpoint selection paradigm, and shows improvement vs. recent baselines.